# Variable Neighborhood Strategy Adaptive Search for Optimal Parameters of SSM-ADC 12 Aluminum Friction Stir Welding

**Suppachai Chainarong [1], Thanatkij Srichok [1],\*, Rapeepan Pitakaso [1] , Worapot Sirirak [2], Surajet Khonjun [1] and Raknoi Akararungruangku [3]**

1. Department of Industrial Engineering, Faculty of Engineering, Ubon Ratchathani University, Ubon Ratchathani 34190, Thailand; suppachai.ch.62@ubu.ac.th (S.C.); rapeepan.p@ubu.ac.th (R.P.); surajet.k@ubu.ac.th (S.K.)
2. Department of Industrial Engineering, Faculty of Engineering, Rajamangala University of Technology Lanna Chiang Rai, Chiang Rai 57120, Thailand; worapotsirirak@rmutl.ac.th
3. Department of Industrial Engineering, Faculty of Engineering, KhonKaen University, KhonKaen 40000, Thailand; raxaka@kku.ac.th
\* Correspondence: thanatkij.s@ubu.ac.th

**Abstract:** In this study, we present a new algorithm for finding the optimal friction stir welding parameters to maximize the tensile strength of a butt joint made of the semisolid material (SSM) ADC 12 aluminum. The welding parameters were rotational speed, welding speed, tool tilt, tool pin profile, and rotational direction. The method presented is a variable neighborhood strategy adaptive search (VaNSAS) approach. The process of finding the optimal friction stir welding parameters comprises five steps: (1) identifying the type and range of friction stir parameters using a literature survey; (2) performing experiments according to (1); (3) constructing a regression model using the response surface method optimizer (RSM optimizer); (4) using VaNSAS to find the optimal parameters for the model obtained from (3); and (5) confirming the results from (4) using the parameter levels obtained from (4) to perform real experiments. The computational results revealed that the tensile strength generated from VaNSAS was 3.67% higher than the tensile strength obtained from the RSM optimizer parameters. The optimal parameters obtained from VaNSAS were a rotation speed of 2200 rpm, a welding speed of 108.34 mm/min, a tool tilt of 1.23 Deg, a tool pin profile of a hexagon, and a rotational direction of clockwise.

**Keywords:** friction stir welding; differential evolution algorithm; D-optimal; SSM-ADC 12

## 1. Introduction

Aluminum alloys are important for building components in various industries that require low-weight and high-strength materials, such as the automotive, marine, and aviation industries. These industries use aluminum alloys and cast aluminum [1] due to their strength, weldability, machinability, corrosion resistance, and formability [2,3]. Today, cast aluminum is formed using a semisolid state process to reduce the defects of cast aluminum. The semisolid metal (SSM) aluminum was created for assembled welding.

The fusion welding process, which uses a high temperature for melting and welding, is difficult with aluminum materials due to problems associated with the thermal expansion coefficient and the low melting point. The weld seam causes metallurgy-related problems, defects, and issues with the mechanical properties, including low strength, distortion, shrinkage, and porosity in weld lines [4–6]. A solid-state welding joint was used to solve the problems associated with weld joint ability and metallurgy. In particular, the friction stir welding process produced a low melting point and good material mixing using a stirred tool [7,8]. Friction stir welding (FSW) does not melt and recast but utilizes plastic deformation at the welding location from frictional heat between the tool and the material. This allows for a good weld line structure [9–11].

A lot of research has been conducted on process parameters and their effects on FSW aluminum alloys. The process parameters for welding are important for control-ling heat generation, plastic deformation, material flow, material mixing, metallurgy, strength, and reducing defects [12–24]. Recent studies on the optimal process parameters for aluminum welding found that rotation speed, welding speed, shoulder diameter per pin diameter ratio, tool geometry, and surface shoulder can cause increases or decreases in the strength and number of weld line defects [25–27]. Several approaches for predicting the optimal aluminum welding parameters (factorial design [28], Taguchi design [29], the response surface method, a combined method [30], etc.) are shown in Table 1.

**Table 1.** Predicted approaches and parameters optimization from literature reviews.

| Authors | Approaches | Materials | Joint Welding | | Optimized Parameters | | | | | | | |
|---|---|---|---|---|---|---|---|---|---|---|---|---|
| | | | Similar | Dissimilar | Rotation Speed | Welding Speed | Tilt Angle | Tool Geometry | D/d Ratio | Axial Force | Tool Material | Rotational Direction |
| This work | Hybrid method D-optimal experimental design and VaNSAS | SSM-ADC 12 | ✔ | | ✔ | ✔ | ✔ | ✔ | | | | ✔ |
| Meengam and Sillapasa (2020) [28] | Factorial design | SSM-Al 6063 | ✔ | | ✔ | ✔ | | ✔ | | | | |
| Srichok et al., 2020 [30] | Combination of RSM and MDE | AA 6061-T6 | ✔ | | ✔ | ✔ | | ✔ | | ✔ | ✔ | |
| Hartl et al., 2020 [31] | Gaussian Process Regression | EN AW 6082-T6 | ✔ | | ✔ | ✔ | | | | | | |
| Prasad and Namala 2018 [32] | Taguchi method and Anova | AA5083 and AA6061 | | ✔ | ✔ | ✔ | ✔ | | | | | |
| Shanayas and Edwin Raja Dhas 2017 [33] | RSM | AA 5052-H32 | ✔ | | ✔ | ✔ | ✔ | ✔ | | | | |
| Kadaganchi et al., 2015 [15] | RSM | AA2014-T6 | ✔ | | ✔ | ✔ | ✔ | ✔ | | | | |
| Hartl et al., 2020 [34] | ANN | AA 6082-T6 | ✔ | | ✔ | ✔ | | | | | | |
| Bayazid et al., 2015 [35] | Taguchi method | AA 6063-7075 | | ✔ | ✔ | ✔ | | | | | | |
| Shojaeefard et al., 2014 [36] | Combination of FEM and ANN | AA 5083 | ✔ | | ✔ | ✔ | | | | | | |
| Teimouri and Baseri 2013 [37] | Combination of ABC and ICA | aluminum | ✔ | ✔ | ✔ | ✔ | | | | | | |
| Roshan et al., 2013 [38] | Combination of RSM, ANFIS and SA | AA 7075 | ✔ | | ✔ | ✔ | | ✔ | | ✔ | | |
| Aydin et al., 2010 [39] | Combination of Taguchi method and GRA | AA 1050 | ✔ | | ✔ | ✔ | | | ✔ | | | |
| Tansel et al., 2010 [40] | Combination of ANN and GA | AA 1080 | ✔ | | ✔ | ✔ | | | | | | |
| Lakshminarayanan and Balasubramanian 2008 [41] | Taguchi method | AA RDE-40 | ✔ | | ✔ | ✔ | | | | ✔ | | |
| Yousif et al., 2008 [42] | ANN | Al alloy | ✔ | | ✔ | ✔ | | | | | | |

RSM = response surface method, MDE = modified differential evolution, FEM = finite element method, ANN = artificial neural networks, SA = simulated annealing algorithm, ANFIS = adaptive neuro-fuzzy inference systems, GRA = grey relation analysis, GA = genetic algorithm, ABC = artificial bee colony algorithm, ICA = imperialistic competitive algorithm.

From a literature review, we found that seven parameters had an influence on the weld line properties. However, several studies only selected the four parameters that most affected the metallurgical characteristics and mechanical properties, i.e., rotation speed, welding speed, tool geometry, and tilt angle [15,43]. The axial force, tool material parameter, and D/d ratio of a tool have an influence on the initial welding but do not affect the weld movement. This is because welding starts with high heat and the plastic deformation of the material. Thereafter, the heat and plastic deformation are controlled by the rotation speed and welding speed parameter. The weld movement reduces the tool wear and axial force [44]. Therefore, both the axial force and tool material parameter were neglected in several previous study (see Table 1). In addition, tool rotational direction parameter was not considered an important factor with commonsense in FSW. Because this parameter does not affect to the thermal generation, and material deformation in welding the similar material. That it was not essential to effect study on weld line property.

However, some research found that the rotational direction parameter has not been shown to directly affect the weld line properties. Contrarily, it was found that tool rotational direction has been interaction effect together with welding direction or tool pin shape especially thread pin that affected to turbulence flow of material, reduced defect, smooth surface and reduced thinning flash deformation [45–49].

Furthermore, the optimization of welding parameters is an important method for controlling the welding. The experimental design can differ notably in terms of the number of experiments and accuracy. Nevertheless, when we compare the experimental design methods with similar factor numbers, the D-optimal design exhibits the lowest number of experiments and produces the highest accuracy [50–52]. For this reason, the D-optimal

design has been used as the experimental design in many studies. Moreover, the optimized parameter approach was developed for experimental data analyses using a single or combined approach for optimal response prediction: a combined approach gives more accurate response predictions than the single approach [38,40].

In addition, several heuristic methods were used for optimization prediction to improve the speed and success of the tour routing, transportation, agriculture, and manufacturing processes [53–58]. Several studies used heuristic methods for optimized problem solving, such as adaptive large neighborhood search (ALNS), genetic algorithm (GA), differential evolution (DE), particle swarm optimization (PSO), variable neighborhood strategy adaptive search (VaNSAS), etc. These heuristics are successful in problem solving and lead to better solutions than conventional methods [15,43,44,50,51]. The VaNSAS algorithm was found to give higher accuracy than other algorithms, with an average solution accuracy of 99.92% [44,51,52,54]. Assembly line balancing problems in manufacturing processes and location routing problems in transportation can be solved using the VaNSAS algorithm to reduce transport time and costs. Therefore, the VaNSAS algorithm is used by researchers for solution optimization.

The optimization of FSW process conditions was a major achievement in the field of optimal condition prediction [30,36]. Aluminum alloys have most often been used in research on FSW, with less research considering SSM aluminum materials.

Therefore, this work focuses on finding the optimized process conditions for FSW. The welding type was butt joint welding for SSM ADC 12 aluminum. The ADC 12 aluminum material is a new material that was developed for use as part of the marine and automotive industries. However, the optimal parameters for FSW using the ADC 12 material remain unknown. Thus, finding the optimized parameters is the first step in the D-optimal design for experimental generation and parameter optimization with the variable neighborhood strategy adaptive search (VaNSAS) approach. The process parameters for optimization conditions were rotation speed, welding speed, tool rotational direction, tool tilt angle, and tool geometry. The tensile strength of the weld line optimal response prediction was analyzed using a scanning electron microscope (SEM) in order to discuss the weld characteristics and defects.

## 2. Literature Review

Finding the optimal process parameters for friction stir welding is important for ensuring a high weld line quality. The type of material affects the process parameters and properties of the weld line. The process conditions of FSW have been found to change the values of the controlling parameters. Parameter optimization is important for the strength and quality of weld lines. Recently, Meengam and Sillapasa found that the parameter optimization based on a local search method and a factorial design method gave the optimum FSW parameters of the SSM-6063 aluminum material [25]. The three parameters in their study were the rotation speed, welding speed, and optimal tensile strength. The geometry tool was a cylindrical pin. The weld line of the optimum welding condition exhibited a high tensile strength of 123.59 MPa. The structure of the weld line area was characterized as being coarse grain and the thermal–mechanical effect zone as equiaxed grain, which decreased in welding strength. The structure exhibited an intermetallic compounds phase: a defect that leads to reduced effective strength of the weld line. Moreover, the variation in welding parameters led to changes in the heat input, i.e., an increase or decrease in the strength, metallurgy, and/or defects in the welding structure [59].

Shanavas and Dhas [33] presented an RSM method for process parameter optimization to improve the ultimate tensile strength and elongation. The optimal process conditions led to the highest tensile strength and fewest defects in a weld line structure. A change in the welding speed influenced the variable strength and welding structure. Prasad and Namala [32] used the Taguchi design and ANOVA to obtain the optimal process parameters of aluminum dissimilar welding. The optimum welding conditions led to good elongation

and an improved working hardness. In addition, the optimum welding conditions ensured good mechanical weld line properties. Tehyo et al. [60] studied the welding parameters for the dissimilar welding of SSM 356 and AA6061-T651. The tensile strength of the weld line was studied under optimal process welding conditions. The influence of the process parameters showed heat-generated change, material flow behavior, and material mixing in the weld line.

In addition, the heuristic method for FSW can be used to achieve satisfactory process parameter optimization solutions. In 2020, Hartl et al. [31] predicted the ultimate tensile strength when welding the AW 6082-T6 material. The highest tensile strength prediction was 255 MPa with optimal welding conditions. The Gaussian process regression algorithm exhibited a prediction accuracy of 96%. Recently, Hartl et al. [34] used an artificial neural network (ANN) to predict the optimal welding conditions for AW-6082-T6. It was found that the ANN algorithm provided an exact solution in 88% of predictions. Yousif et al. [42] used the ANN algorithm for FSW aluminum prediction of tensile stress, bending stress, and elongation of the weld line. The rotation speed and transverse speed were the study parameters. The predicted average error of the ANN algorithm was 0.84% for tensile stress, 7.6% for bending stress, and 13.29% for elongation.

Palani et al. [61] and Suenger et al. [62] used a D-optimal design based on RSM to determine the optimal tensile strength and hardness in friction stir welding. They found that the increase in the hardness of the welded seam depended on controlling the welding temperature. The optimal welding conditions kept the temperature stable and encouraged a good structure in the weld seam, which led to optimal tensile strength and hardness. The prediction model had an average accuracy of 97.30%.

The predicted process parameter optimization using the local search method and heuristic method from the literature review demonstrated an increased predicted errors as compared to using a single approach. Therefore, a combined approach is recommended for increased accuracy. Srichok et al. [30] presented the optimization of friction stir welding AA6061-T6 with the combined RSM and modified differential evolution (MDE) algorithm. The RSM was used to identify factors that influence tensile strength and the MDE for optimized factor prediction. The four optimized factors were rotation speed, welding speed, axial force tool pin geometry, and tool material. The optimal process parameters gave the highest ultimate tensile strength of 95.10% for the base material due to the homogeneity and lack of defects. This method provided a prediction accuracy of 98.52%. Accordingly, Roshan et al. [38] presented a welding method for AA 7075 combining RSM, ANFIS, and SA for the optimization of the ultimate tensile strength and hardness. The main process factors were the rotation speed, welding speed, axial force, and tool pin. The optimum conditions led to the highest ultimate tensile strength, but the structure exhibited a small defect in the weld line.

In 2014, Shojaeefard et al. [36] optimized the friction stir welding process using the finite element method (FEM) and ANN approach. The first step used the FEM and the second step used the ANN algorithm for optimized response prediction. The welding led to increased heat generation with an increased rotation speed and decreased welding speed. Moreover, the high welding temperature reduced the tool axial force, which acts to increase the tool life. The weld line displayed a high efficiency of 91%. Tansel et al. [40] used the GONNS algorithm, based on the ANN-GA combined approach, for parameter optimization. The process factors were rotation speed and welding speed. The mechanical properties analyzed by the GONNS algorithm were tensile strength, elongation, and hardness, and the GONNS algorithm exhibited an accuracy of 97.4% for mechanical property prediction.

The literature review revealed that the combined method had an efficiency of 96% for predictions [63]. The single-step method gave a 6–32% difference in range in terms of tensile strength, but the combined method demonstrated a 1–6%, difference in tensile strength, which was lower than the single-step method, as shown in Table 2. However, an accurate prediction depends on the heuristic algorithm specifically. The several approaches in-volved in one heuristic algorithm can lead to a high prediction accuracy. The VaNSAS

algorithm method is a heuristic algorithm used to accurately predict a solution. The problem-solving accuracy of the VaNSAS algorithm has been proven by several re-searchers. Jirasirilerd et al. [57] and Pitakaso et al. [58,64] used the VaNSAS algorithm for production and planning problem solving. The operating algorithm used in the VaNSAS process can be the differential evolution algorithm, the iterated local search, the swap method, the modified differential evolution algorithm, the large neighborhood search, or the shortest processing time-swap. The VaNSAS algorithm has a high solution prediction accuracy of up to 99.99%. Therefore, the VaNSAS algorithm is suitable for optimal parameter prediction in the FSW process. The most important factors influencing the weld line quality include rotation speed and welding speed, which affect the heat generation and microstructure. The tool geometry and tool tilt angle are also important factors for controlling in-process welding and can increase the effectiveness of the weld line and decrease the number of defects. These four factors should not be ignored in the FSW process.

**Table 2.** Effective comparing of tensile strength between single and hybrid method.

| Method | Materials | Optimal Parameters | | | | | | | Tensile Strength (MPa) | | % Difference of Tensile Strength |
| | | Rotation Speeds (rpm) | Welding Speed (mm/mim) | Tilt Angle (°) | Tool Pin Geometry | D/d Ratio | Axial Force (kN) | Tool Material | Weldline | Base Material or Prediction | |
| --- | --- | --- | --- | --- | --- | --- | --- | --- | --- | --- | --- |
| Single | SSM-6063 [28] | 1320 | 60 | 3 | cylindrical | 3.84 | - | H13 tool steel | 120.7 | 149 | 18.99 |
| | EN AW-6082-T6 [31] | 1700 | 1500 | 2 | conical thread and three flats | - | - | SK 50 | 255 | 332.97 | 23.41 |
| | AA 2099-T83 [59] | 800 | 450 | 1.5 | tapered triangular and thread | - | 15 | H13 tool steel | 390 | 558 | 30.1 |
| | AA5052-H32 [60] | 600 | 65 | 1.5 | tapered square pin | - | - | H13 tool steel | 202.58 | 216.58 | 6.47 |
| | SSM 356-AA6061-T651 | 2000 | 80 | 3 | cylindrical | 4 | 4.4 | JIS-SKH 57 tool steel | 197.1 | 290 of AA6061 | 32.06 of AA6061 |
| Hybrid | AA6061-T6 [30] | 1417 | 60.21 | - | Hexagonal-taper | - | 8.44 | SKD11 | 294.84 | 310 | 4.89 |
| | AA7075 [38] | 1400 | 105 | - | Square | - | 7.5 | High cabon | 227 | 241 | 5.80 |
| | AA 1080 [40] | 500 | 6.25 | - | - | - | - | - | 112 | 115 | 2.60 |
| | Aluminum alloy [38] | 509.35 | 10.10 | - | Straight cylindrical | - | 7 | high carbonic steel | 110.26 | 112 | 1.15 |

## 3. Materials and Methods

### 3.1. Identifying the Number of Parameters of Interest and Their Ranges and Levels

In this study, we surveyed the literature to find the number and level of each parameter that affects the maximum tensile strength.

In our experiment, we used the information in Tables 1 and 2 to determine the parameters and their values. The minimum and maximum of the parameters were set as follows: (1) rotational speed: 1100 to 2200 rpm; (2) welding speed: 80 to 200 mm/min; and (3) tool tilt angle: 0 to 6 Deg. We used two types of tool pin profile: cylindrical or hexagonal, and different rotations: clockwise (CW) or counterclockwise (CCW). The combinations of parameters for the experiment are detailed in Figure 1.

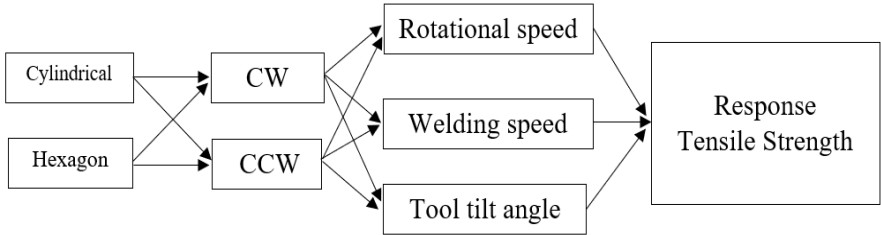

**Figure 1.** Parameters used in the experiment.

### 3.2. Using D-Optimal Experimental Design to Find the Regression Model of the Parameters for Friction Stir Welding

The experiments were carried out according to a D-optimal experimental design with five factor parameters, and the levels used in the experiments were continuous and categorical, as shown in Table 3. In this case, a D-optimal experimental design was used to select h design points from those set by the embedding algorithm, resulting in 19 minimum

model points, five points for estimation of the lack-of-fit, and five points for replicates. Finally, an experimental plan with a total of 29 points was created.

**Table 3.** Parameters in the experiment.

| Continuous Variable | | |
|---|---|---|
| **Parameter** | **Levels** | |
| | **−1** | **1** |
| Rotation speed (rpm), S | 1100 | 2200 |
| Welding speed (mm/min), F | 80 | 200 |
| Tool tilt angle Deg., T | 0 | 6 |
| **Categorical Variables** | | |
| **Parameter** | **Levels** | |
| Tool pin profile, P | Cylindrical | Hexagon |
| Rotational direction, M | Clockwise: CW | Counterclockwise: CCW |

The upper and lower limits of the parameters in the statistical Design-Expert software (Stat-Ease, Inc., Minneapolis, MN, USA) were set to −1 and 1.

The intermediate coded values were calculated using Equation (1):

$$Original = (Scaled\ [(X_{Max} + X_{min})] + X_{Max} + X_{min})/2 \tag{1}$$

where *Scaled* is the required coded value of a variable *X*; *X* is any value of the variable from $X_{min}$ to $X_{Max}$; and $X_{min}$ and $X_{Max}$ are the lowest and highest predefined values of the parameter, respectively. Table 3 provides the details of each coded and uncoded parameter, which includes the upper and lower bounds of these parameters.

D-optimal software was used to design and create the experimental models and problem analysis. The quadratic model shown in Equation (2) is expected to be obtained from the experiment:

$$y = b_0 + \sum_i^k b_i x_i + \sum_i^k b_{ii} x_i^2 + \sum_i \sum_j b_{ij} x_i x_j + \varepsilon \tag{2}$$

where *y* is the maximum tensile strength (response), $x_i$ is the uncoded levels of the variables, $\varepsilon$ is the fitting error, the coefficient $b_0$ is the constant value or intercept and coefficients, and $b_i$, $b_{ii}$, and $b_{ij}$ represent the linear, quadratic, and interaction terms of the variables, respectively [65].

### 3.3. Using Variable Neighborhood Strategy Adaptive Search to Find the Optimal Parameters

The variable neighborhood strategy adaptive search (VaNSAS) approach is a new metaheuristic for searching for a solution in a large area. Several approaches were gathered into one method in a black box for searching for the best solution. The structure of the VaNSAS algorithm is easy to understand. Jirasirilerd et al. [57] used the VaNSAS algorithm for problem solving, including the five steps.

#### 3.3.1. Generate a Set of Tracks

This step is for the generation of an initial solution. A set of 3 × 1 tracks is randomly generated. The number of randomly generated tracks is fixed. The random value that is inserted into the positions of the track is bounded by the upper and lower value of each parameter. Position 1 of the track represents the rotational speed value, while Positions 2 and 3 represent the welding speed and tool tilt angle, respectively. A track can consist of five elements. An example of the five-track system is provided in Table 4. Each element represents a parameter, as shown in Table 4.

**Table 4.** The track and the element value.

| Track Number | 1 | 2 | 3 | 4 | 5 |
|---|---|---|---|---|---|
| **Element** | | | | | |
| 1 (Rotational speed) | 0.36 | 0.74 | 0.41 | 0.63 | 0.62 |
| 2 (Welding speed) | 0.71 | 0.32 | 0.03 | 0.80 | 0.29 |
| 3 (Tool tilt angle) | 0.20 | 0.03 | 0.12 | 0.19 | 0.18 |

The values of the elements shown in Table 4 can be decoded into a solution ac-cording to the track transforming process (TTP). The TTP can be explained as follows:

The rotation speed, welding speed, and tool tilt angle values are shown in Table 3. Each element has a value, e.g., the value of Track 1 in Element 1 (rotational speed) is 0.36. The value of the real rotation speed is $1100 + [(0.36)(2200 - 1100)] = 1496$. Other values in the element are decoded by the mechanism shown above, using Equation (3):

$$R = L + e(U - L) \tag{3}$$

where $R$ denotes the real value of the parameter, $L$ denotes the lowest value of the parameter that is allowed, $U$ is the highest value of the parameter that is allowed, and $e$ is the value in position. An example of transforming to the real parameter value is shown in Table 5.

**Table 5.** Track transforming process.

| Factor | Track | | | | |
|---|---|---|---|---|---|
| | 1 | 2 | 3 | 4 | 5 |
| 1 (Rotational speed) | 1496 | 1914 | 1551 | 1793 | 1782 |
| 2 (Welding speed) | 165.2 | 118.4 | 83.6 | 176 | 114.8 |
| 3 (Tool tilt angle) | 1.2 | 0.18 | 0.72 | 1.14 | 1.08 |

The values shown in Table 5 for each track are used for the calculations from Equation (3).

### 3.3.2. Perform Track Touring Process in a Specified Black Box

All tracks select a black box to improve the quality of the current solution. The black boxes that are used in this section are as follows: (1) the differential evolution algorithm; (2) the swap method; and (3) the insertion method. We used roulette wheel selection to select the track and probability to select the black box, which is controlled by Equation (4):

$$P_{bt} = \frac{FN_{bt-1} + (1 - F)A_{bt-1} + KI_{bt-1}}{\sum_{bt=1}^{n} W_{bt}} \tag{4}$$

where $P_{bt}$ is the probability of the selection of a black box in iteration $t$; $N_{bt-1}$ is the number of tracks that have selected a black box in the previous iteration; $A_{bt-1}$ is the average objective value of all the tracks that selected a black box in the previous iteration; and $I_{bt-1}$ is a reward value, increased by 1 if a black box finds the best solution in the last iteration, but set to 0 if this is not the case. Additionally, $W_{bt}$ is the weight of the black box, $F$ is the scaling factor ($F = 0.5$), and $K$ is the parameter factor ($K = 0.3$).

There are three black boxes used in this article: (1) the differential evolution algo-rithm; (2) the swap method; and (3) the insertion method, which are explained in Section 3.3.3.

### 3.3.3. Black Box Operation

The black box operation step identifies a solution out of the three selected approaches in each black box. These are as follows:

Simplify Differential Evolution Algorithm (SDE)

The SDE is a global algorithm composed of five steps: (1) generating an initial solution; (2) performing a mutation process; (3) performing a recombination process; (4) performing a selection process; and (5) repeating Steps (2) to (4) until the termination condition is met. Details of the algorithm are as follows:

Step 1: Initial population.

Randomly select two tracks from the track that do not select SDE as the black box.

Step 2: Perform a mutation process by applying Equation (5) as the mutation formula:

$$DE/rand/1 \; : \; V_{i,G+1} = X_{r1,G} + F(X_{r2,G} - X_{r3,G}) \tag{5}$$

where $X_{i,G}$ is the target vector; $V_{i,G+1}$ is the mutant vector; and $X_{r1G}, X_{r2G}, \; and X_{r3G}$ are the vectors that we randomly selected from the target vector. $F$ is a scaling factor that is a self-adaptive parameter ranging from 0 to 2. In our experiment, $F$ was initially set to 0.8 and randomly changed to reduce or increase by 0.05 in each individual vector. The value of F of the best vector in the current iteration was set to the current $F$ value, which was used as the base $F$ value to be adapted by the vectors in the next iteration.

Step 3: Perform the recombination process.

In this step, the mutant vector is transformed to a trial vector using Equation (6), where is the trial vector, $rand_{i,j}$ is a random number, and CR is set to 0.6.

$$U_{i,G} = \begin{cases} V_{i,j,G} \text{ if } rand_{i,j} \; \leq \text{CR} \\ X_{i,j,G} \text{ if } rand_{i,j} \; > \text{CR} \end{cases} \tag{6}$$

Step 4: Perform the selection process.

This step is used to find the next-generation target vector. The new target vector can be obtained by using Equation (7):

$$X_{i,G+1} = \begin{cases} U_{i,G}, \text{ if } f(U_{i,G+1}) \leq f(X_{i,G}) \\ X_{i,G} \text{ ,otherwise} \end{cases} . \tag{7}$$

Step 5: Repeat Steps (1) to (4) until the termination condition is met.

In this step, we use the number of iterations as the termination condition and the maximum number of iterations is set to 100.

K-Exchange Method (KEM)

The swap method is a simple heuristic normally used to improve the solution quality. It is composed of six steps: (1) randomly generating the K value, which is an integer and can be 1 or 2; (2) randomly selecting a track (not the current track) that selects KEM as the black box; (3) randomly selecting k points to swap for values in the elements; (4) swapping the values in the positions of the elements; (5) if the objective function is better than the current solution, update the value of the element, and (6) repeating Steps (1) to (5) until the termination condition is met (the number of iterations is set to 100). An example of the K-exchange method is shown in Appendix A.

The current solution that selects KEM as the black box is Track Number 1 and a track that does not select KEM as the black box is randomly chosen; we chose Track Number 2. K is randomly selected to be 2, and the two randomly selected positions are Positions 3 and 1. The result of the swap is shown in Appendix A. If the new Track Number 1 is better than the old one, the new track will be used as the current solution for the next iterations.

K-Transition Method (KTM)

KTM is a simple heuristic composed of six steps: (1) randomly pick a value of K that lies between 1 and 2 (2) randomly select a track (not the current track) that selects the K-Swap method, (3) randomly select k points to transit the value of the element, (4) randomly generate a new value for the position; (5) if the objective function is better than the current solution, update the value of the element, and (6) repeat Steps (1) to (5)

until the termination condition is met (the number of iterations is set to 100). An example of the K-transition method is shown in Appendix B.

The current solution that selects KTM as the black box is Track Number 1. Random numbers are used for the transit values. The value of K is set to 2 and Positions 2 and 3 are selected to be transited. The new track for Track 1 is shown in Appendix B.

The next step of VaNSAS is to update the track, as is explained in the following section.

### 3.3.4. Update the Track

The value in position of the track will be updated using Equation (8):

$$Z_{ijt+1} = Z_{ijt} + \alpha \left( Z_{ijt}^{pb} - Z_{ijt} \right) + (1 - \alpha) \left( Z_{ijt}^{gb} - Z_{ijt} \right) + \beta \left( Z_{2jt} - Z_{3jt} \right), \tag{8}$$

where $Z_{ijt+1}$ denotes the value of track $i$, element $j$, and iteration $t + 1$, respectively. Additionally, $\alpha$ and $\beta$ are random numbers with a value of 0 to 1, $Z_{ijt}$ is track $i$, element $j$ is the last iteration of the black box, $Z_{2jt}$ is the first randomly selected track, $Z_{3jt}$ is the second randomly selected track, $Z_{ijt}^{pb}$ is the personal best track, and $Z_{ijt}^{gb}$ is the global best solution.

### 3.3.5. Repeat the Steps

Repeat the steps in Sections 3.3.2–3.3.4 until the termination condition is met. The stopping criterion here is the maximum number of iterations, which is set to 1000 (resulting from the preliminary test).

The pseudocode of VaNSAS used in this paper is shown in Appendix C.

### 3.4. The Methods Compared

In this section, two well-known metaheuristics are compared with the proposed meth-od. These are: (1) the differential evolution algorithm (DE) and (2) the genetic algo-rithm (GA).

### 3.4.1. Differential Evolution Algorithm (DE)

We modified the DE proposed by Srichok et al. [30]. It is composed of four general steps: (1) generating an initial solution; (2) performing a mutation process; (3) performing a recombination process; and (4) performing a selection process. The DE used in our experiment is shown in Appendix D.

### 3.4.2. Genetic Algorithm (GA)

A genetic algorithm (GA) is a nature-inspired metaheuristic composed of four steps: (1) generating an initial solution; (2) performing a mutation procedure; (3) performing a crossover procedure; and (4) performing a selection procedure. We modified the GA proposed by Metchell and Melanie [66] to use in our problem. The pseudo code of GA used in our research is shown in Appendix E.

## 4. Experimental Framework and Results

The computational results are divided into three parts: the result from the D-optimal experimental design, the result for the proposed problem using VaNSAS, and the result of the real experiment using the parameter levels determined in the second part to confirm the reliability of the theoretical levels of the parameters.

### 4.1. Optimization Process by D-Optimal Experimental Design

We designed the experiment using the Design-Expert software. Five controlled parameters were set: rotational speed (S), welding speed (F), tool tilt angle (T), tool pin profile, and type. We used different rotations. Details of the specimens are listed in Table 6.

**Table 6.** Details of the tested specimens.

| Material | Size (mm) | Thickness (mm) | Ultimate Tensile Strength (MPa) |
|---|---|---|---|
| (SSM)ADC 12 | 75 × 150 | 6 | 208.53 |

The D-optimal experimental design produced 29 experiments, as shown in Table 7; thus, 29 specimens were prepared, as shown in Figure 2.

**Table 7.** Actual design of experiments.

| Run | Rotation Speed | Welding Speed | Tool Tilt Angle(Deg) | Tool Pin Profile | Rotational Direction | Tensile Strength (MPa) |
|---|---|---|---|---|---|---|
| 1 | 2062.92 | 142.75 | 3.41 | Hexagon | ccw | 96.28 |
| 2 | 1110.00 | 80.00 | 6.00 | Hexagon | ccw | 140.38 |
| 3 | 2023.17 | 168.30 | 4.03 | Cylindrical | ccw | 99.03 |
| 4 | 1803.75 | 200.00 | 6.00 | Cylindrical | cw | 91.02 |
| 5 | 1110.00 | 80.00 | 0.00 | Cylindrical | ccw | 43.65 |
| 6 | 2220.00 | 80.00 | 6.00 | Cylindrical | cw | 151.23 |
| 7 | 1110.00 | 80.00 | 6.00 | Hexagon | ccw | 134.11 |
| 8 | 1371.09 | 151.66 | 2.51 | Cylindrical | ccw | 53.49 |
| 9 | 1110.00 | 200.00 | 6.00 | Hexagon | cw | 137.95 |
| 10 | 1654.93 | 148.96 | 3.67 | Hexagon | cw | 166.65 |
| 11 | 2216.02 | 95.19 | 1.34 | Cylindrical | ccw | 44.85 |
| 12 | 1110.00 | 200.00 | 6.00 | Hexagon | cw | 131.38 |
| 13 | 1484.65 | 96.41 | 2.72 | Hexagon | cw | 159.49 |
| 14 | 2220.00 | 80.00 | 6.00 | Cylindrical | cw | 153.76 |
| 15 | 1705.76 | 134.87 | 2.91 | Hexagon | cw | 150.98 |
| 16 | 1715.48 | 141.78 | 3.18 | Hexagon | ccw | 126.76 |
| 17 | 1827.64 | 128.46 | 1.71 | Cylindrical | cw | 171.87 |
| 18 | 1110.00 | 80.00 | 0.00 | Cylindrical | ccw | 40.52 |
| 19 | 1395.44 | 112.16 | 3.28 | Hexagon | cw | 143.6 |
| 20 | 1307.14 | 164.53 | 2.04 | Cylindrical | cw | 155.18 |
| 21 | 1338.46 | 200.00 | 1.61 | Cylindrical | ccw | 49.51 |
| 22 | 1896.10 | 168.39 | 3.46 | Hexagon | cw | 162.96 |
| 23 | 1688.31 | 152.64 | 4.57 | Hexagon | ccw | 111.02 |
| 24 | 1893.30 | 167.26 | 1.61 | Cylindrical | cw | 130.02 |
| 25 | 1445.26 | 142.86 | 3.42 | Cylindrical | ccw | 51.59 |
| 26 | 1863.64 | 147.26 | 3.56 | Hexagon | ccw | 101.48 |
| 27 | 1391.97 | 143.76 | 4.03 | Cylindrical | cw | 121.32 |
| 28 | 1717.21 | 140.79 | 1.22 | Cylindrical | cw | 168.45 |
| 29 | 2062.92 | 142.75 | 3.41 | Hexagon | ccw | 97.46 |

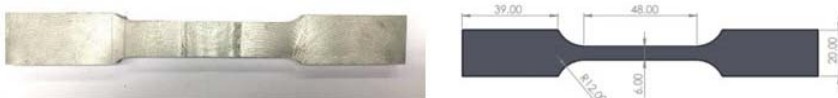

**Figure 2.** An example of the prepared specimens.

After the welding process was finished, the tensile strength was measured using a tensile test machine (model NRI-TS501-300, Narin Instruments Co., Ltd., Bangmueng Mueng, Samutprakarn, Thailand; Figure 3). The welded specimens were tested until broken; then, the tensile strength was recorded, as shown in Table 7.

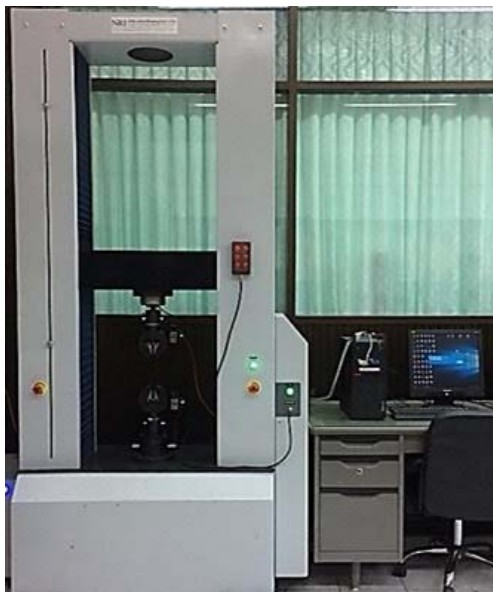

**Figure 3.** The tensile test machine.

Table 7 provides the experimental results; the maximum tensile strength of the workpiece was obtained from Experiment Number 17, which used a rotational speed of 1827.64 rpm, a welding speed of 128.46 mm/min, a tool tilt angle of 1.71 Deg, a cylindrical pin profile, and clockwise rotation. This welding parameter produces the highest tensile strength of 171.87 MPa.

On the basis of the experimental results in Table 7, the D-optimal experimental design software was used to produce a regression equation to show the relationship between the variable values, and four models were formulated, which are defined in Table 8. It was found that the model forms were accepted as mathematical models because the *p*-values of both equations were less than 0.05. The mathematical model had a coefficient of decision making ($R^2$) from the influence of variables equal to 95.26% and the revised coefficient (adjusted $R^2$) was greater than 86.73%, which confirms that the regression model obtained the right format.

**Table 8.** The ANOVA results for tensile strength response using the Design-Expert software.

| Source of Variation | Sum of Squares | DF | Mean Squares | F-Value | *p*-Value |
|---|---|---|---|---|---|
| Model | 48,619.12 | 18 | 2701.06 | 11.17 | 0.0002 |
| Linear | 13,684.2 | 5 | 2736.83 | 11.31 | 0.001 |
| Square | 174.9 | 3 | 58.31 | 0.24 | 0.866 |
| Interaction | 8516.4 | 10 | 851.64 | 3.52 | 0.030 |
| Residual Error | 2418.8 | 10 | 241.88 | | |
| Lack-of-Fit | 2368.79 | 5 | 473.76 | 47.34 | 0.0003 |
| Pure Error | 50.0 | 5 | 10.01 | | |
| Total | 51,037.9 | 28 | | | |
| R-sq = 95.26% R-sq(adj) = 86.73% | | | | | |

Four models were formulated from the data using Design-Expert software:

$$CW_{\_Cylindrical} = 318 + 0.081\,S - 2.46\,F - 9.6T - 0.000055\,S*S + 0.0065\,F*F + 1.56\,t*t$$
$$+ 0.000464\,S*F + 0.0060\,S*T - 0.162\,F*T \tag{9}$$

$$CCW\_Cylindrical = 78 + 0.130\,S - 2.43\,F + 18.9\,T - 0.000055\,S*S + 0.0065\,F*F + 1.56\,T*T$$
$$+ 0.000464\,S*F + 0.0060\,S*T - 0.162\,F*T \tag{10}$$

$$CW\_Hexagon = 306 + 0.030\,S - 1.33\,F - 22.7\,T - 0.000055\,S*S + 0.0065\,F*F + 1.56\,T*T$$
$$+0.000464\,S*F + 0.0060\,S*T - 0.162\,F*T \tag{11}$$

$$CCW\_Hexagon = 82 + 0.079\,S - 1.30\,F + 5.8\,T - 0.000055\,S*S + 0.0065\,F*F + 1.56\,T*T$$
$$+0.000464\,S*F + 0.0060\,S*T - 0.162\,F*T \tag{12}$$

where $S$, $F$, and $T$ represent the rotational speed, welding speed, and tool tilt angle, respectively. The D-optimal experimental design software was used to determine the optimal solution using the regression model in Equations (9)–(12), 200.13 MPa. The parameters generating this solution were as follows: a rotational speed of 1374.07 rpm, a welding speed of 167.68 mm/min, a tool tilt angle of 0.10 Deg, a cylindrical pin profile, and clockwise rotation, as shown in Figure 4.

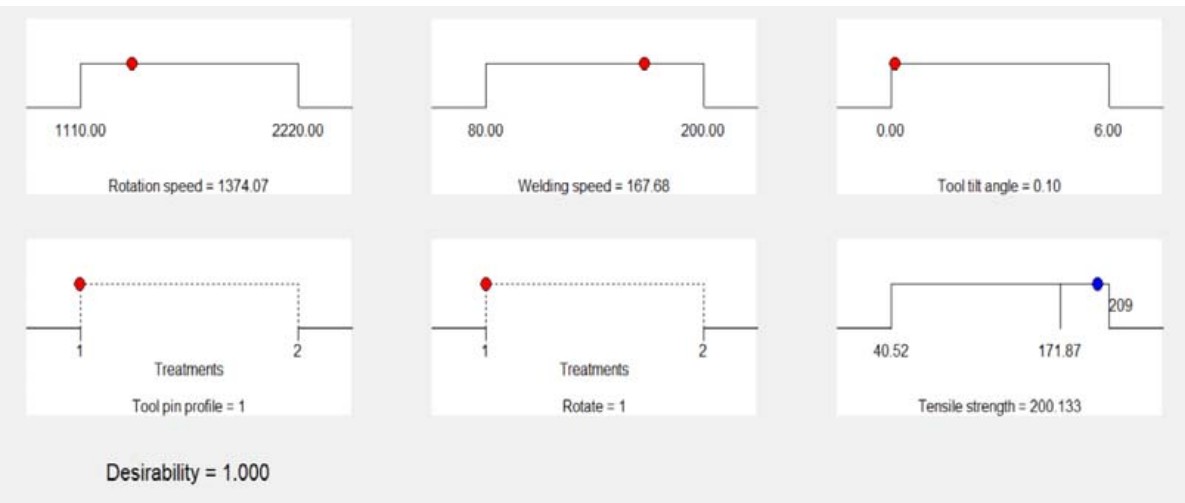

**Figure 4.** D-optimal experimental design predicted model using optimal conditions.

### 4.2. Results Using Variable Neighborhood Strategy Adaptive Search (VaNSAS)

The proposed VaNSAS was coded in PyCharm (JetBrains Americas, Inc., Marlton, NJ, USA) using a PC with an Intel Core i7 3.70 GHz CPU and 8 GB DDR4 RAM. The objective function of the model was given by the D-optimal experimental design (Equations (9) to (12)) and used in VaNSAS to find the optimal solution of the problem subject to Equations (13) to (15). The parameters value ranges that can be applied using the D-optimal experimental design software are shown in Table 3. The maximum tensile strength range was not limited because it was the response from the input parameters that we aimed to determine.

$$1100\text{ rpm} \leq S \leq 2200\text{ rpm} \tag{13}$$

$$80\text{ mm/min} \leq F \leq 200\text{ mm/min} \tag{14}$$

$$0\text{ degrees} \leq T \leq 6\text{ degrees} \tag{15}$$

In this study, VaNSAS is self-adaptive, as explained in Section 3.3. In our experiment, the maximum number of iterations was set to 1000.

VaNSAS was applied to increase the effectiveness of finding the optimal value parameters. Each method was executed 30 times and the best tensile strength was determined as shown in Table 9.

Table 9 lists the results of solutions using DE, GA, and VaNSAS to find the most suitable parameters for friction stir welding. The maximum tensile strength of 207.79 MPa was obtained using VaNSAS. The computational times of DE, GA, and VaNSAS were 10.58, 10.95 and 10.23 min respectively. With the results from the statistical test, we used ANOVA to check if the computational time of each heuristic was different. We found that

the *p*-value was 0.618. This demonstrates that the computational times for each heuristic did not significantly differ. Therefore, VaNSAS recorded the same computational time as the other methods, but it provided a better solution.

**Table 9.** The computational results of the maximum tensile strength of each heuristic.

| Type of Rotational Direction/Tool Pin Profile | Output Values of Each Heuristic Tensile Strength | | | | | |
|---|---|---|---|---|---|---|
| | DE | | GA | | VaNSAS | |
| | Tensile (MPa) | Com (s) | Tensile (MPa) | Com (min) | Tensile (MPa) | Com (min) |
| CW_Cylindrical | 205.99 | 10.8 | 205.98 | 11.2 | 206.0 | 11.2 |
| CW_Hexagon | 206.97 | 10.2 | 205.90 | 10.4 | 207.79 | 10.4 |
| CCW_Cylindrical | 204.53 | 11.8 | 202.16 | 9.8 | 206.53 | 9.6 |
| CCW_Hexagon | 204.99 | 9.5 | 204.94 | 12.4 | 205.97 | 9.9 |

Remark: Com is computational time of a heuristic.

The parameter/type values that generated the maximum tensile strength values are shown in Table 10.

**Table 10.** Output values of VaNSAS with respect to input process parameters.

| Condition | | Unit | Result |
|---|---|---|---|
| | Rotational speed | Rpm | 2200 |
| | Welding speed | mm/min | 108.34 |
| | Tool tilt | Deg | 1.23 |
| Optimal parameter | Pin profile | Hexagon | |
| | Rotational direction | CW | |
| | Maximum tensile strength | MPa | 207.79 |

### 4.3. Verifying the Results by Testing Optimal Parameters with Actual Specimens

After we obtained the appropriate parameters from VaNSAS, as shown in Table 10, we performed a test and compared the results of confirmed experiments with the calculated VaNSAS results in order to check if the parameter values generated by VaNSAS could form a welded material with the maximum tensile strength. Twelve replications were conducted and the average maximum tensile strength was recorded, as shown in Table 11.

$$\%diff = \frac{tensile\ strength\ ^{Exp} - tensile\ strength\ ^{VaNSAS}}{tensile\ strength^{Exp}} \times 100\% \tag{16}$$

where *tensile strength* $^{Exp}$ is the maximum tensile strength generated by the real experiment and *tensile strength*$^{VaNSAS}$ is the maximum tensile strength generated by VaNSAS. The average maximum tensile strength obtained from the confirmed experiment was $206.85 \pm 0.886$ MPa, which is close to the calculated VaNSAS result of 207.79 MPa. The tensile strength comparison of both the confirmed experiment and the VaNSAS prediction, using Equation (16), exhibited a percentage difference of 0.45%. The tensile strength comparison between the confirmed experiment and the base material specimen revealed a percentage difference of 0.80%, as shown in Table 12. Moreover, the comparisons of tensile strength of each approach and the base material specimen are shown in Table 13. The VaNSAS prediction had the lowest percentage difference of tensile strength, i.e., 0.35%, and the accuracy was 99.65%, as compared with the base material specimen.

**Table 11.** Comparison of the experimental and the VaNSAS results.

| Variable Parameter | Unit | Result | Tensile Strength (Mpa) | | % Difference |
| --- | --- | --- | --- | --- | --- |
| | | | Confirmed Experiment | VaNSAS | |
| Rotational speed | rpm | 2200 | | | |
| Welding speed | mm/min | 108.34 | | | |
| Tool tilt | Deg | 1.23 | 206.85 ± 0.886 | 207.79 | 1.93 |
| Pin profile | | Hexagon | | | |
| Rotational direction | | Clockwise | | | |

**Table 12.** Different percentage comparison of tensile strength.

| Method | Tensile Strength (MPa) | % Diff |
| --- | --- | --- |
| Base material specimen | 208.53 | - |
| Initial experiment | 171.87 | 17.58 |
| D-Optimal prediction | 200.13 | 4.02 |
| VaNSAS prediction | 207.79 | 0.35 |
| Confirmed experiment | 206.85 | 0.80 |

**Table 13.** Different percentage comparison of method.

| Method | % Tensile Strength Difference of Method |
| --- | --- |
| Initial experiment vs. D-Optimal | 14.12 |
| Initial experiment vs. VaNSAS | 17.28 |
| Initial experiment vs. confirmed experiment | 16.91 |
| VaNSAS | 3.68 |
| D-Optimal vs. confirmed experiment | 3.24 |
| VaNSAS vs. confirmed experiment | 0.45 |

*4.4. The Reliability and Effectiveness Testing of the Proposed Methods*

In this step, we assessed the reliability and effectiveness of the proposed method, as suggest in [67–70].

We used 12 test examples collected from the literature [15,28,33,71–79]. We identified the optimal parameters using the proposed methods and compared these with the results obtained using the D-optimal/RSM approaches on the aforementioned literature. The results are shown in Tables 14 and 15. Table 14 is the computational result of the proposed method test with the standard test examples from the literature, while Table 15 is the statistical test (*p*-value) using Wilcoxon sign rank test.

We can conclude that VaNSAS outperformed the other methods. It significantly improved the solution quality by 0.33 to 2.61%, as compared to the other methods. As regards the time it took for each method to find the best solution (maximum tensile strength), we can see that VaNSAS was able to identify a 100% improved solution as compared with the other methods. Therefore, we can conclude that the VaNSAS significantly improved the solution quality as compared with the other methods.

**Table 14.** Result of the reliability test.

| Instances | Authors | Ultimate Tensile Strength/Tensile Strength (MPa) | | | |
|---|---|---|---|---|---|
| | | D-Optimal/RSM | GA | DE | VaNSAS |
| 1 | Meengam and Sillapasa [28] | 120.7 | 123.55 | 124.82 | 125.11 |
| 2 | Jenarthanan et al. [71] | 105.47 | 106 | 107.63 | 108.02 |
| 3 | Tanmoy Medhi et al. [72] | 129.73 | 132.22 | 134.25 | 135.06 |
| 4 | Shanavas and Edwin raja dhas [33] | 202.58 | 204 | 206.86 | 206.54 |
| 5 | Ramanjaneyulu et al. [15] | 445 | 448.51 | 452.22 | 454.12 |
| 6 | Farzad et al. [73] | 535.5 | 536.24 | 538.62 | 540.08 |
| 7 | Masoud Ahmadnia et al. [74] | 187.35 | 210.53 | 211.24 | 212.54 |
| 8 | Ravi Sankar and Umamaheswarrao [75] | 184 | 186.35 | 187.39 | 189.32 |
| 9 | Hridya Nand Singh et al. [76] | 236 | 238.45 | 239.62 | 240.06 |
| 10 | Amit Goyal and Ramesh Kumar Garg [77] | 253.4 | 255.43 | 258.91 | 260.20 |
| 11 | JANNET et al. [78] | 288 | 288.54 | 290.78 | 290.98 |
| 12 | Kavitha et al. [79] | 211.48 | 211.95 | 212.97 | 213.08 |

**Table 15.** Statistical test using Wilcoxon sign rank test (*p*-value).

| | GA | DE | VaNSAS |
|---|---|---|---|
| D-optimal | 0.002 | 0.002 | 0.002 |
| GA | | 0.002 | 0.002 |
| DE | | | 0.005 |

*4.5. Microstructure Analysis*

The experimental welding condition exhibited different weld line microstructures and tensile properties in the FSW of SSM ADC 12 aluminum. The differences in structure and tensile properties were analyzed by microstructure photography from confirmed experiments and the highest tensile strength value is given in Table 10. Figure 5 shows the microstructure characteristics in BM, TMAZ-AS, TMAZ-RS, and SZ from the experimental welding test with the VaNSAS optimal conditions, i.e., rotation speed: 2200 rpm, welding speed: 108.34 mm/min, tilt angle: 1.23 Deg, a hexagonal pin tool, and a clockwise direction. We found that SZ had no defects and was soundly welded. In the BM, β-Al5FeSi compounds demonstrated shape transformations, from platelike flakes to small flakes, as shown in Figure 5a,c. In TMAZ-AS, the β-Al5FeSi compounds were found to be arranged according to the direction of rotation and the edge of the pin tool due to the friction force; this caused the β-Al5FeSi compounds to fracture, leading to them being dragged in the direction of rotation as shown in Figure 5b. Similarly, in TMAZ-RS, we found that the β-Al5FeSi compounds transformed from platelike shapes into rod shapes, with smaller particle sizes, as shown in Figure 5d. The small intermetallic phase of β-Al5FeSi and the dispersion in the weld line structure led to an in-creased tensile strength. As a result of the precipitation of the intermetallic phase, β-Al5FeSi is a strengthening mechanism that protects the slip plane from the tensile force. Therefore, the optimal welding weld line condition gives the highest tensile strength according to Meengam and Sillapasa [25].

The optimized welding conditions for the D-optimal prediction were a rotational speed of 1374.07 rpm, a welding speed of 167.68 mm/min, a tool tilt angle of 0.10 Deg, a cylindrical pin profile, and a clockwise rotation. The microstructure characteristics resulting from these welding conditions were as follows: microvoids, zigzag line defects, and kissing bond defects in SZ and TMAZ, as shown in Figure 6. As a result, the tool geometry lacked an angular shape, which caused turbulence and negatively affected the material and mixing in weld joint, according report of Elangovan and Bal-asubramanian [19]. Therefore, the tensile resistance of the weld joint exhibited a lower performance than the weld joint resulting from the optimal welding parameters as predicted by VaNSAS.

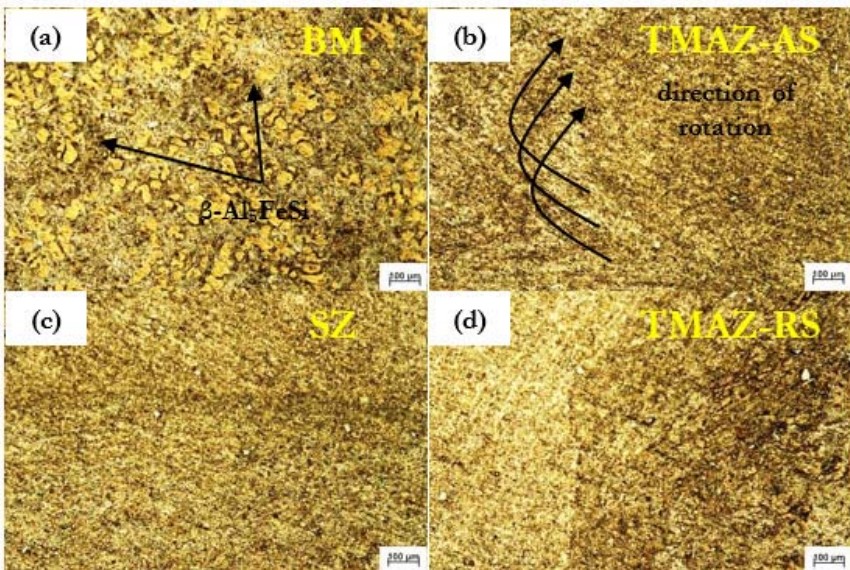

**Figure 5.** Characteristics of microstructure (rotation speed: 2200 rpm, welding speed: 108.34 mm/min, tilt angle: 1.23 Deg, hexagonal pin tool, and clockwise direction) in: (**a**) BM, (**b**) TMAZ-AS, (**c**) SZ, and (**d**) TMAZ-RS.

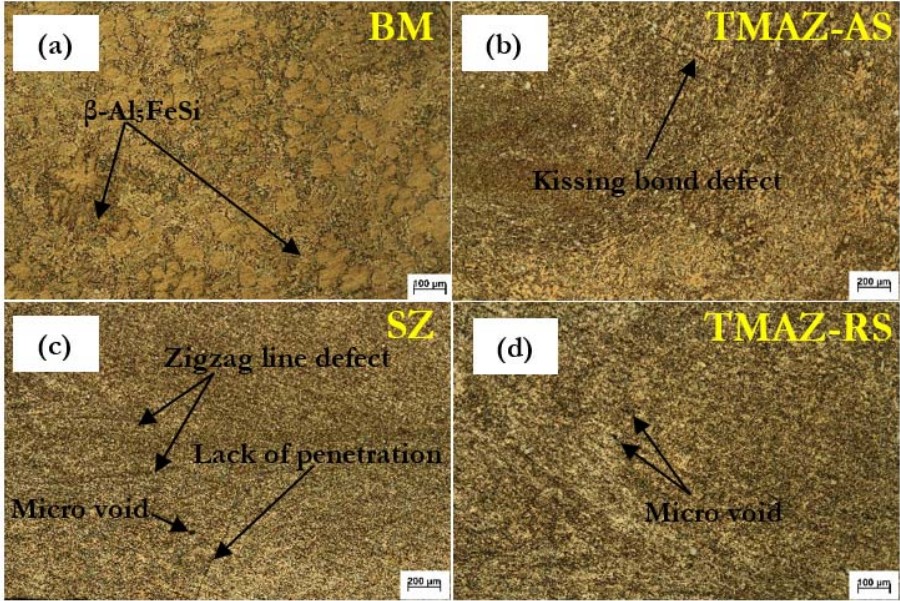

**Figure 6.** Characteristics of microstructure (rotational speed of 1374.07 rpm, welding speed of 167.68 mm/min, tool tilt angle of 0.10 Deg, cylindrical pin profile, and clockwise rotation) in: (**a**) BM, (**b**) TMAZ-AS, (**c**) SZ, and (**d**) TMAZ-RS.

However, the unoptimized factors for FSW in the initial experiment using the SSM ADC 12 aluminum alloy resulted in poor mechanical properties, because a rotation speed of 1827.64 rpm, a welding speed of 128.46 mm/min, a tilt angle of 1.71 Deg, a cylindrical tool, and a clockwise direction caused defects. The formation of a large void (defect) in SZ is shown in Figure 7c. This was due to insufficient material flow from the RS side to the AS side, which connects to TMAZ-AS, resulting in a very low tensile strength (see Figure 7b). It is worth noting that β-Al5FeSi compounds in TMAZ-RS have a larger particle size because of the lower heat input, which contributes to complete precipitation, as shown in Figure 7d. BM found that the characteristics of β-Al5FeSi compounds were similar to those in previous experiments, as shown in Figure 7a.

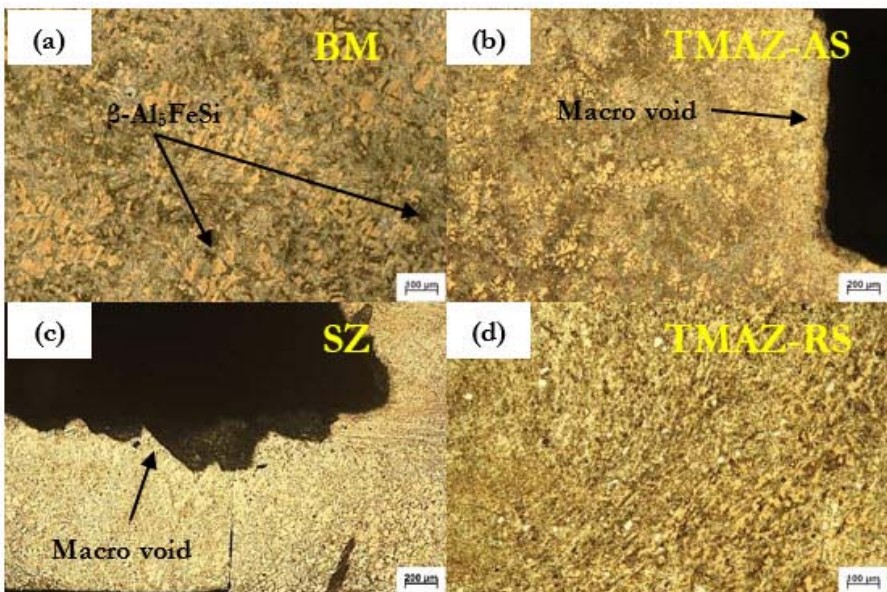

**Figure 7.** Characteristics of microstructure (rotation speed: 1827.64 rpm, welding speed: 128.46 mm/min, tilt angle: 1.71 Deg, cylindrical tool, and clockwise direction) in: (**a**) BM, (**b**) TMAZ-AS, (**c**) SZ, and (**d**) TMAZ-RS.

Figure 8 shows the distribution of $\beta$-$Al_5FeSi$ compounds in the specimens resulting from a rotation speed of 2200 rpm, a welding speed of 108.34 mm/min, a tilt angle of 1.23 Deg, a hexagonal pin tool, and a clockwise direction, evaluated using SEM at 12,000×. The $\beta$-$Al_5FeSi$ compounds, whose shapes and particle sizes were assessed, were newly crystallized and distributed by phase due to friction force or thermal stress. The $\beta$-$Al_5FeSi$ compounds in SZ, which formed from the $\beta$-phase in the $\beta'$-phase formation of $\beta'$-$Mg_2Si$ and possibly $Cu_2Mg_8Si_6Al_5$ compounds, are shown in Figure 8c–e. TMAZ-RS and TMAZ-RS found that an $\beta'$-$Mg_2Si$ compound was cracked in the intergranular area because the SSM ADC 12 aluminum alloy contained silicon (Si) and iron (Fe), causing brittle properties, as shown in Figure 8a,b. Although friction force or thermal stress affect the occurrence of cracks, the analysis showed that the area at the bottom of the joint (the end of tool pin area) generated heat during FSW, but this resulted in no cracks, as shown in Figure 8e,f. Cracks in the intergranular area led to tearing when subjected to static force, as dynamic force could not be applied to adjust the tensile properties. The size of $\beta'$-$Mg_2Si$ compounds is around 3–10 μm in SZ and around 10–17 μm in TMAZ-RS and TMAZ-RS. The difference in the size of $\beta'$-$Mg_2Si$ compounds was caused by different experimental factors.

Various experiments did not demonstrate a positive effect on the metallurgical structure. The conditions were as follows: rotation speed: 1827.64 rpm, welding speed: 128.46 mm/min, tilt angle: 1.71 Deg, a cylindrical tool, and a clockwise direction. We noted incomplete welds and a microvoid, as shown in Figure 9. It is worth noting that the areas that received less heat during FSW generated $\beta'$-$Mg_2Si$ compounds with a needle-shaped microstructure of around 16–22 μm, as shown in Figure 9a,b,e,f. For SZ under the shoulder tool (see Figure 9c), we found that $\beta'$-$Mg_2Si$ compounds exhibited a flakelike structure and were around 3–11 μm. Similarly, in the middle of the weld, we found that the size of $\beta'$-$Mg_2Si$ compounds was around 8–15 μm, as shown in Figure 9d. When we compared them with the flake-shaped $\beta'$-$Mg_2Si$ compounds, we noted a higher dynamic force than that produced by a needle-shaped microstructure. Therefore, the needle-shaped $\beta'$-$Mg_2Si$ were small, rounded, and distributed throughout the SZ, contributing to the good tensile properties.

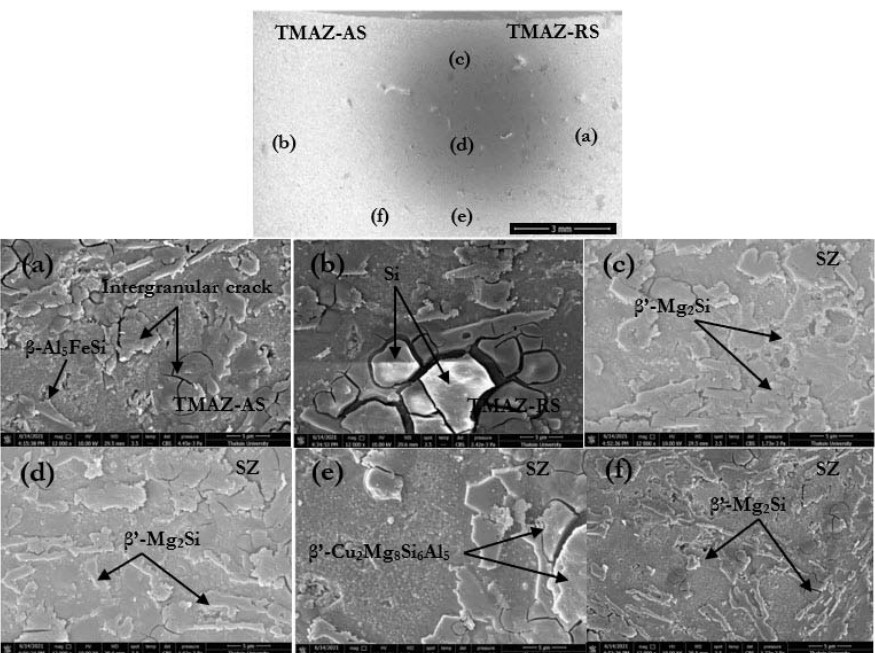

**Figure 8.** The microstructures with SEM with rotation speed of 2200 rpm, welding speed of 108.34 mm/min, tilt angle of 1.23 Deg, hexagonal pin tool, and clockwise direction. (**a**) TMAZ-RS, (**b**) TMAZ-AS, (**c**) Top-SZ, (**d**) Middle-SZ, (**e**) Bottom-SZ, (**f**) Side-SZ.

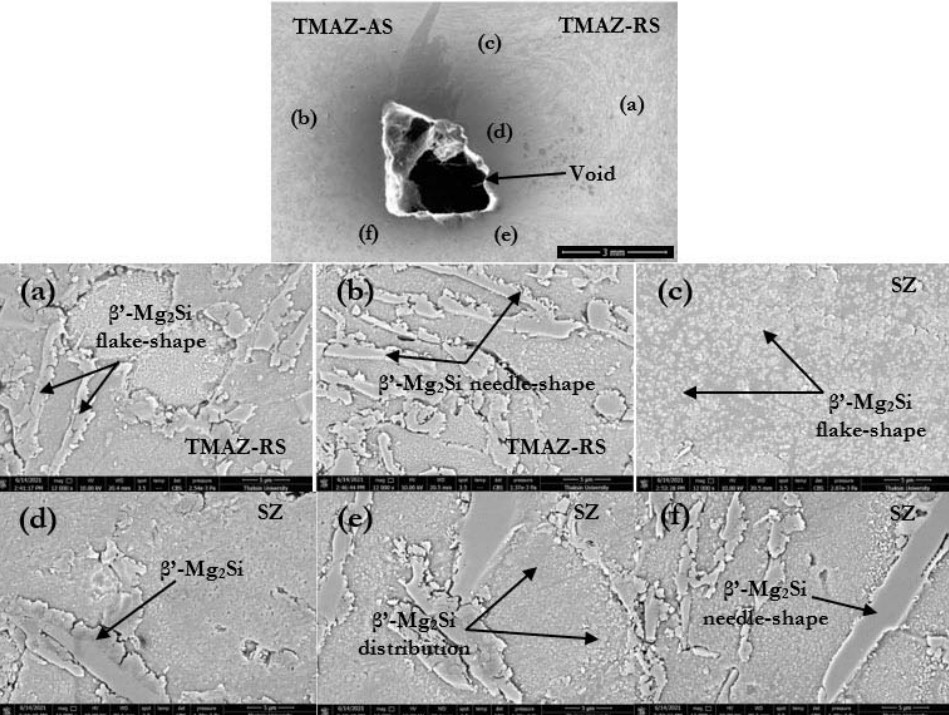

**Figure 9.** The microstructures with SEM with rotation speed of 1827.64 rpm, welding speed of 128.46 mm/min, tilt angle of 1.71 Deg, cylindrical tool, and clockwise direction. (**a**) TMAZ-RS, (**b**) TMAZ-AS, (**c**) Top-SZ, (**d**) Middle-SZ, (**e**) Bottom-SZ, (**f**) Side-SZ.

## 5. Conclusions

In this study, we aimed to identify the optimal parameters of FSW for producing the best tensile strength. The D-optimal approach was used for the experimental design, and the VaNSAS algorithm method was used for optimal parameter prediction. The heuristic

operation in the black box of the VaNSAS algorithm included the differential evolution algorithm, the swap method, and the insertion method for finding a solution. The welding parameters were rotation speed, welding speed, tool tilt angle, tool pin profile, and tool rotational direction. The evaluation of ADC 12 aluminum welding from FSW led us to draw the following conclusions:

Experiments confirmed that the optimal welding conditions with the VaNSAS algorithm are as follows: rotation speed: 2200 rpm, welding speed: 108.34 mm/min, tilt angle: 1.23 Deg, a hexagonal pin tool, and a clockwise direction. These conditions produced a tensile strength of 206.85 MPa. The structure of the weld line exhibited no defects and the precipitation of small intermetallic phases, i.e., β-Al5FeSi, β′-Mg2Si, and Cu2Mg8Si6Al5, with a uniform distribution indicated strength. Therefore, the tensile strength depends on the heat generation of the welding parameter. The influence of the heat input affects the success of FSW. Sufficient heat input leads to a complete microstructure without defects and this is generated by optimal welding conditions. The tensile property of the weld line gives a high tensile value and vice versa. Insufficient heat input and the appearance of defects indicate unsuitable welding conditions and reduce the tensile strength of the weld line.

The D-optimal approach, which was used for the experimental design, and the VaNSAS algorithm, which was used for optimization, gave a lower prediction of tensile strength by 0.35% as compared with that of the material specimen. The VaNSAS algorithm was able to identify the best solution for optimal parameter welding. Therefore, the hybrid method prediction involving VaNSAS provided a precise solution with 99.65% accuracy. However, the percentage difference in tensile strength when comparing the VaNSAS algorithm and the confirmed experiments was 0.45%. The VaNSAS algorithm prediction gave the best value for optimal parameter welding. Therefore, the hybrid method prediction involving VaNSAS indicated a precise solution with 99.20% accuracy.

Future work should be focused on generating optimal parameter welding for multi-objective responses. Tensile strength hardness, bending strength, and toughness are important mechanical properties for the weld line using the FSW process.

**Author Contributions:** Conceptualization, S.C. and T.S.; methodology, T.S. and R.P.; validation, R.P. and W.S.; writing—original draft preparation, S.C. and S.K.; writing—review and editing, S.C. and W.S.; project administration, T.S. and R.A. All authors have read and agreed to the published version of the manuscript.

**Funding:** This research received no external funding.

**Institutional Review Board Statement:** Not applicable.

**Informed Consent Statement:** Not applicable.

**Data Availability Statement:** Data is contained within the article.

**Acknowledgments:** The authors would like to thank Department of Industrial Engineering, Faculty of Engineering, Ubon Ratchathani University in Thailand.

**Conflicts of Interest:** The authors declare no conflict of interest.

## Appendix A

**Table A1.** Example of KEM.

| Track Number | Current Track Track 1 | Randomly Selected Track Track 2 | New Track Track 1 |
|---|---|---|---|
| Element | | | |
| 1 (Rotational speed) | 0.36 | 0.74 | 0.74 |
| 2 (Welding speed) | 0.71 | 0.32 | 0.71 |
| 3 (Tool tilt angle) | 0.20 | 0.03 | 0.03 |

## Appendix B

**Table A2.** Example of KTM.

| Track Number | Current Track Track 1 (1) | Random Number (2) | New Track Track 1 (3) |
|---|---|---|---|
| Element | | | |
| 1 (Rotational speed) | 0.36 | 0.45 | 0.36 |
| 2 (Welding speed) | 0.71 | 0.89 | 0.89 |
| 3 (Tool tilt angle) | 0.20 | 0.14 | 0.14 |

## Appendix C. Pseudocode of VaNSAS

| **Algorithm A1:** Variable Neighborhood Strategy Adaptive Search (VaNSAS) |
|---|
| **input:** *Number of Track (NP), Problem Size (D), Mutation Rate (F), Recombination rate (R), Number of Black box (NBB)* |
| **output:** Best_Track_Solution |
| **begin** |
|     Population = Initialize **Track** (NP, D) |
|     IBPop = Initialize **Information B**B(**NB**B) |
|     encode Population **as a track** |
|     **while** *the stopping criterion is not met **do*** |
|         *for i = 1: NP* |
|             *Set u [j] = randomnumber)_[j]* |
|             *//selected black box by RouletteWheelSelection* |
|             *selected_BB = RouletteWheelSelection(BBPop) using Equation (3)* |
|             *If(selected_BB = 1) Then* |
|             *new_u = SDE (u)* |
|             *Else if(selected_BB = 2)* |
|             *new_u = K-exchange (u)* |
|             *Else if(selected_BB = 3)* |
|             *new_u = K_Transition (u)* |
|             *IF(CostFunction(new_u) ≤ CostFunction($V_i$)) Then* |
|             *$V_i$ = new_u* |
|             *//Loop for update heuristics information of Intelligence box* |
|             ***For j = 1: NBB*** |
|             *$BBPop_i$ using Equaltion (8)* |
|             ***End For Loop****//end update heuristics information* |
|             ***End For Loop*** |
|     End |
|     Decode WP to get the solution for the problem |
|     Return Best track Solution |
| **end** |

## Appendix D. Pseudocode of DE

| **Algorithm A2:** Differential evolution algorithm (DE) |
|---|
| **input:** *Population size (NP), Problem Size (D), Mutation Rate (F), Recombination rate (R)* |
| **output:** Best_Vector_Solution |
| **begin** |
|     Population = Initialize Population (NP, D) |
|     encode Population to WP |
|     **while** *the stopping criterion is not met **do*** |

*for i = 1: NP*
    $V_{rand1}$, $V_{rand2}$, $V_{rand3}$ = Select_Random_Vector (WP)
    **For** *j = 1: D//Loop for the mutation operator*
      $V_y [j] = V_{rand1} [j] + F (V_{rand2} [j] + V_{rand3} [j])$
    **End For Loop**//*end mutation operator*
    **For** *j = 1: D//Loop for recombination operation*
      *If (rand$_j$ [0,1) < R) Then*
        $u [j] = V_i [j]$
      *Else*
        $u [j] = V_y [j]$
      **End For Loop**//*end recombination operation*
    *IF(CostFunction(u) $\leq$ CostFunction(V$_i$)) Then*
      $V_i = u$
    **End For Loop**
End
    decode WP to get the solution for the problem
    Return Best Vector Solution
end

## Appendix E. Pseudocode of GA

**Algorithm A3:** Genetic Algorithm (GA)

**input:** *Population Size (NP), Problem Size (D), Mutation Rate (M), Crossover Rate (CR)*
**output:** Best_Vector_Solution
**begin**
    Population = Initialize Population (NP, D)
    encode Population to WP
    **while** *the stopping criterion is not met* **do**
      *parents = WP*
      *for i = 1: NP//Loop for crossover operation*
        **For** *j = 1: D*
        *If(rand$_j$ [0,1) < CR ) Then*
          *offspring$_i$ [j] = parents$_i$ [j]*
          *offspring$_{i+1}$ [j] = parents$_{i+1}$ [j]*
        *Else*
          *offspring$_i$ [j] = parents$_{i+1}$ [j]*
          *offspring$_{i+1}$ [j] = parents$_i$ [j]*
        **End For Loop**
      **End For Loop**//*end crossover operation*
      *for i = 1: NP//Loop for mutation operation*
        **For** *j = 1: D*
        *If(rand$_j$ [0,1) < M ) Then*
          *Mutation(offspring$_i$ [j])*
          **End For Loop**
        **End For Loop**//*end mutation operation*
        //Add the child population to the parent population
        NWP = stack(*parents, offspring*)
        wp_size = length(NWP)//Set number of new population
        *for i = 1: wp_size//Loop for evaluate operation*
          *cost_ scores $_{i+1}$= CostFunction(NWP$_{i+1}$)*
        **End For Loop**//*end evaluate operation*
        *//selection operation*
        new_wp = *Sorted*(new_ population, *cost_scores*)
        *WP = NWP [1:NP]*
        decode WP to get the solution for the problem
        Return Best Vector Solutionend
end

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
