# Peer review of "Variable Neighborhood Strategy Adaptive Search for Optimal Parameters of SSM-ADC 12 Aluminum Friction Stir Welding"

_processes, doi:10.3390/pr9101805_

Round 1

Reviewer 1 Report

In this paper, the authors present a new algorithm for finding the optimal parameters of friction 14 stir welding to maximize the tensile strength of a butt joint made of the semi-solid material (SSM) 15 ADC12 aluminum. This work is of great interest for the researchers working on the semi-solid metal (SSM) aluminum. However, this work must be carefully revised before being accepted for publication on Processes.

Changes which must be made before publication:

  1. The figures quality should be improved, such as Figure 2 and Figure 3, as well as the figures and tables should be numbered consecutively in accordance with their appearance in the text, for instance, there are two Tables numbered Table 9 while there is no Table 13.

  1. The consumed time should also be presented and compared for the proposed method and other approaches.

  1. The microstructure of tensile fracture for different cases should also be presented and compared.

Author Response

Thank you very much for your suggestion. 

Reviewer 2 Report

Without the comments.

Author Response

(The authors gave the same response as above.)

Reviewer 3 Report

The manuscript presented a methodology to find optimal process parameters of FSW for SSM-ADC12 material, combining experiment design and optimization methods. As the authors stated, determining the optimal FSW process for SSM-ADC12 is of public interest and can be a contribution to the community.

It is commented as follow:

1 The title of this manuscript might be a little bit lengthy; it is suggest to make it concise if possible.

2 For the keywords, it is suggested to add the material name as well; readers rely on the keywords while searching papers.

3 Adding "rotational direction" of the tool into process parameters needs more supporting evidence at least. The authors stated that "the rotational direction parameter has not been shown to affect the weld line properties"; it is easy to understand that, if the joints are symmetric to the welding line (which is the case in this manuscript), the rotational direction doesn't either affect the thermal field nor the stress field, even in the initial stage. If there is no indication suggesting "rotational direction" has an impact on properties of the joints, then it shouldn't be included in the investigation.

4 Page 14, Table 9, it might be best to use "Ultimate Tensile strength", instead of "Maximum Tensile strength" (UTS).

5 Page 17 Section 4.3: while verifying the VaNSAS' prediction experimentally, 12 specimens are tested and the average value is shown in Table 12; it might be better to add the variations in the same time.

6 It is noticed from Page 14 Table 8 that UTSs of the same process parameters vary slightly (NO worry; it is expected); for example, Run 5 and Run 18 share the same process parameters, while UTSs differ around 3 MPa. While claiming the prediction of VaNSAS is better than D-Optimal, it will be more convincing if the difference is much more than variance of FSW experiment.

Author Response

(The authors gave the same response as above.)

Reviewer 4 Report

  1. It is suggested to rewrite the section 3 part, it is to much now espcially for the methods part. Maybe the author can use “appendix” to inlculde them.
  2. How the author judge reliability of your results? The the data set is very small. so limited data for five variables? I don't think it's reasonable.

Author Response

(The authors gave the same response as above.)

Round 2

Reviewer 1 Report

I believe the manuscript has been sufficiently improved to warrant publication in Processes.

Author Response

Thank you very much for the comment. 

Reviewer 3 Report

It is appreciated that the authors responsed very quickly to the comments. After reviewing the revised version, there are raising concerns regarding to the previous comment 3 ("Adding "rotational direction" of the tool into process parameters needs more supporting evidence at least.").

While responsing to the comment 3, the author cited reference 45 "Effect of weld curvature radius and tool rotation direction on joint microstructure in friction stir welding casting alloys", which is really inapprociate and NOT supporting the authors' statement.

As previously stated clearly, "it is easy to understand that, if the joints are symmetric to the welding line (which is the case in this manuscript), the rotational direction doesn't either affect the thermal field nor the stress field, even in the initial stage." But the citation is actually about curved welds (page 125, Fig.1 of the reference 45); in addition, the material is different (3522 AlSi in the reference 45). 

So adding the "rotational direction" into investigation in this manuscript is not scientific, based on common sense and supporting evidences, even misleading to readers. While not denying totally the value to this work, it is strongly recommended to revise the whole methodology.

Author Response

 Thank you very much for the comment. 

Reviewer 4 Report

Dear Editor,

I am satisfied with the responese from the authors.

Author Response

Thank you very much for the comment. 
